# Constitutive auxin response in *Physcomitrella* reveals complex interactions between Aux/IAA and ARF proteins

Meirav Lavy, Michael J Prigge, Sibo Tao, Stephanie Shain, April Kuo, Kerstin Kirchsteiger, Mark Estelle*

Section of Cell and Developmental Biology, Howard Hughes Medical Institute, University of California, San Diego, San Diego, United States

**Abstract** The coordinated action of the auxin-sensitive Aux/IAA transcriptional repressors and ARF transcription factors produces complex gene-regulatory networks in plants. Despite their importance, our knowledge of these two protein families is largely based on analysis of stabilized forms of the Aux/IAAs, and studies of a subgroup of ARFs that function as transcriptional activators. To understand how auxin regulates gene expression we generated a *Physcomitrella patens* line that completely lacks Aux/IAAs. Loss of the repressors causes massive changes in transcription with misregulation of over a third of the annotated genes. Further, we find that the *aux/iaa* mutant is blind to auxin indicating that auxin regulation of transcription occurs exclusively through Aux/IAA function. We used the *aux/iaa* mutant as a simplified platform for studies of ARF function and demonstrate that repressing ARFs regulate auxin-induced genes and fine-tune their expression. Further the repressing ARFs coordinate gene induction jointly with activating ARFs and the Aux/IAAs.

*For correspondence: mestelle@ucsd.edu

**Competing interests:** The authors declare that no competing interests exist.

## Introduction

The plant hormone auxin plays a central role in plant growth and development. Depending on the context, the cellular response to auxin can be very different, including changes in cell division, cell expansion, and differentiation. The hormone acts by regulating the transcription of auxin responsive genes. Strikingly, auxin-regulated gene sets can vary significantly between different cell types consistent with cell specific cellular responses (*Bargmann et al., 2013*). In the absence of auxin, transcription of auxin-regulated genes is repressed by members of a family of repressors called Auxin/INDOLE-3-ACETIC-ACID (Aux/IAA) proteins. The Aux/IAAs are recruited to promoters through an interaction with AUXIN RESPONSE FACTOR (ARF) transcription factors. Repression is relieved when auxin binds to a co-receptor complex consisting of a TRANSPORT INHIBITOR RESISTANT 1/AUXIN F-BOX (TIR1/AFB) F-box protein and an Aux/IAA protein. The Aux/IAA protein is degraded, permitting ARF-dependent transcription to occur [(*Calderon Villalobos et al., 2012*; *Dharmasiri et al., 2005a*; *2005b*; *Kepinski and Leyser, 2005*; *Tan et al., 2007*) reviewed in (*Salehin et al., 2015*; *Wang and Estelle, 2014*)]. The elucidation of the auxin co-receptor mechanism provided the molecular link between auxin perception at the cellular level and subsequent changes in gene expression. However, the mechanisms by which the interactions between TIR1/AFBs, ARFs and Aux/IAAs results in the induction of specific gene sets, as well as the establishment of gene regulatory networks, are not known.

**eLife digest** Auxin is a plant hormone that regulates many aspects of growth and development. It does so by promoting the degradation of proteins called the Aux/IAAs. These proteins normally act to keep genes switched off by interacting with another family of proteins called ARFs that can bind directly to the genes. Some ARFs activate genes, while others repress gene activity. In most plants, large families of genes encode Aux/IAA and ARF proteins and individual members often have overlapping roles, which makes it more difficult to study how they work.

To avoid this problem, Lavy et al. chose to study how these proteins work in a moss called *Physcomitrella patens*, which only has three genes that encode Aux/IAA proteins. The experiments show that the loss of all three Aux/IAA proteins results in the plants becoming completely insensitive to auxin. Furthermore, over a third of known moss genes had altered activity in the mutant plants compared to normal moss. These findings suggest that the Aux/IAA proteins have a bigger role in regulating the activities of genes than previously thought.

Further experiments show that repressive ARFs act by directly competing with the activating ARFs for binding to auxin-regulated genes. However, repression of gene activity by the ARFs is weaker than the effect of the Aux/IAAs. Unexpectedly, the loss of repressive ARFs causes moss plants to become less sensitive to auxin, which Lavy et al. suggest is due to the recruitment of additional Aux/IAA proteins to the genes. Thus these findings demonstrate that control of gene activity by auxin involves the coordinated action of both types of ARFs and the Aux/IAAs. Future challenges are to find out which genes directly bind ARFs and the Aux/IAAs, and to use computational approaches to create models of how auxin regulates gene activity in the moss.

The Aux/IAAs contain, in most cases, three domains: an N terminal repression domain required for the recruitment of the co-repressor TOPLESS (TPL) (*Szemenyei et al., 2008*) domain II, required for interaction with the TIR1/AFB co-receptors; and a C terminal region (designated domains III and IV) that forms a Phox and Bem1 (PB1) domain (*Guilfoyle and Hagen, 2012*).

ARF proteins are generally comprised of an N terminal B3-type DNA binding domain, a variable middle region (MR), and a C terminal PB1 domain. The ARFs have been characterized as activating or repressing based on their behavior in transient protoplast assays (*Guilfoyle and Hagen, 2007*; *Tiwari et al., 2003*; *Ulmasov et al., 1999*). It was recently shown that Arabidopsis ARF5, an activating ARF, interacts with SWI/SNF chromatin remodeling ATPases through its MR, and that this interaction mediates changes in chromatin required for gene activation (*Wu et al., 2015*). On the other hand the repressing activity of the putative repressing ARFs has not been demonstrated in plants. Similar, the mechanism of repression is unknown.

ARF-Aux/IAA interactions occur through their PB1 domains. In addition, the PB1 domain permits homo- and heterodimerization both within and between the Aux/IAA and ARF families. Recent structural studies revealed that ARFs can form higher order ARF and Aux/IAA protein complexes through both their DNA binding domain [ARF-ARF dimerization (*Boer et al., 2014*)] and the acidic and basic faces of their PB1 domains [ARF-ARF and ARF-Aux/IAA multimerization (*Dinesh et al., 2015*; *Han et al., 2014*; *Korasick et al., 2014*; *Nanao et al., 2014*) reviewed in (*Korasick et al., 2015*; *Wright and Nemhauser, 2015*)]. This combinatorial diversity may result in complex regulation of gene expression. Further, tightly regulated negative feedback loops in which the *Aux/IAA* genes are regulated by the ARFs, present an additional layer of complexity.

The body plan of the early-diverged moss *Physcomitrella patens (P. patens)* is relatively simple. The vegetative gametophyte is composed of only a few cell types and developmental stages. Auxin was shown to have a central role in the vegetative growth of *P. patens* (*Prigge and Bezanilla, 2010*). In the filamentous protonemal stage, auxin promotes the differentiation of chloroplast-rich filaments called chloronemata into elongated filaments with fewer chloroplasts called caulonemata (*Ashton et al., 1979*). The development of leafy gametophores is also affected by auxin including stem elongation (*Eklund et al., 2010*; *Fujita et al., 2008*), elongation of the gametophore leaves (*Bennett et al., 2014*; *Decker et al., 2006*), formation of rhizoids from gametophore epidermal cells (*Ashton et al., 1979*), and gametophore branching (*Coudert et al., 2015*). We previously

demonstrated that the molecular mechanism of auxin signaling is conserved between *P. patens* and Arabidopsis (*Lavy et al., 2012*; *Prigge et al., 2010*). This finding, together with its morphological simplicity establishes *P. patens* as a powerful model for studies of auxin signaling.

Here we utilized a *P. patens aux/iaa* null mutant to determine the effects of the complete loss of Aux/IAA-based transcriptional repression. Our analysis reveals that the Aux/IAAs are essential for auxin regulation of transcription indicating that at least in moss, auxin does not affect transcription independently of the Aux/IAAs. Complete loss of Aux/IAA repression dramatically alters the transcriptome, including expression of a large number of genes that are not affected by auxin treatment. In addition, our studies revealed new features of repressing ARF function. We demonstrate that auxin induction of gene expression is controlled by complex interactions between the Aux/IAAs and both activating and repressing ARFs.

## Results

### Loss of the Aux/IAA repressors result in an auxin-constitutive phenotype

Flowering plants possess large families of *Aux/IAA* genes (29 in Arabidopsis). Genetic studies of these genes have relied almost entirely on gain-of-function mutations in the degron motif (*Prigge et al., 2010*; *Mockaitis and Estelle, 2008*). These dominant mutations prevent interaction between the Aux/IAA and the TIR1/AFB co-receptors resulting in stabilization of the affected Aux/IAA and reduced sensitivity to auxin (*Figure 1G*). Very few loss-of-function mutants have been described in flowering plants, presumably because of gene redundancy and a plant completely lacking the entire *Aux/IAA* family is not available. In a recent study, the single *Aux/IAA* gene in the early diverging Liverwort *Marchantia polymorpha (M. polymorpha)* was knocked down using an artificial microRNA(*Flores-Sandoval et al., 2015*). The resulting lines were auxin hypersensitive. However, because these plants retained some auxin responsiveness, there are unlikely to be nulls. The genome of *P. patens* encodes only three *Aux/IAA* genes: *IAA1A*, *IAA1B*, and *IAA2.* To study the role of these genes in auxin response and plant development, and to simplify the study of auxin response mechanisms, we generated a mutant lacking all three genes. To generate the triple mutant, we replaced the *IAA1B* coding region with a β-*glucuronidase (GUS)* marker gene, followed by deletion of the *IAA1A* and *IAA2* coding regions ($P_{IAA1B}$:GUS aux/iaaΔ) (*Figure 1A* and *Figure 1—figure supplement 1*). Because expression of the *IAA1B* gene is induced by auxin (*Lavy et al., 2012*; *Prigge et al., 2010*), we could use the $P_{IAA1B}$:GUS gene as an auxin reporter. Following auxin treatment, very low levels of GUS staining was detected in the $P_{IAA1B}$:GUS line, whereas GUS staining was high in the $P_{IAA1B}$:GUS aux/iaaΔ line both in the absence and presence of applied auxin, indicating that this line displays a constitutive auxin response (*Figure 1B*).

The resulting *aux/iaa* triple knockout mutant (*aux/iaaΔ*) displayed a phenotype that was similar to that of WT plants grown on high levels of auxin for one month, but more extreme (*Figure 1C*). When grown on minimal medium (BCD), WT plants grew leafy gametophores (*Figure 1D*-left panel), while in the presence of auxin, these plants produced gametophores consisting of shoot structures with ectopic brown-pigmented rhizoids and without leaves (*Figure 1D*-right panel). The *aux/iaaΔ* mutant produced gametophores with ectopic rhizoids and was completely insensitive to auxin (*Figure 1C*). To characterize plants at the filamentous protonemal stage, we compared WT, a highly auxin-resistant *IAA2* degron mutant (*iaa2-P328S*), and the *aux/iaaΔ* mutant. The plants were grown for one month on medium supplemented with ammonium tartrate to allow for slower protonemata differentiation (BCDAT medium) (*Figure 1E–H*). Under these conditions, WT plants responded to exogenous auxin by developing long caulonemal filaments (*Figure 1E,F*). Whereas the differentiation of the *iaa2-P328S* auxin-resistant mutant arrested at the primary chloronemal stage (*Figure 1G*), *aux/iaaΔ* plants contained disorganized brown-pigmented filaments (*Figure 1H*). Interestingly, the phenotype of *aux/iaaΔ* plants was highly variable. Each plant developed heterogeneous filaments with varying levels of chloroplasts and brown pigment (*Figure 1H*, left and right panel illustrates the range of phenotypes). Since the *aux/iaaΔ* mutant displayed a phenotype similar to that of auxin treated plants, we also tested its response to reduced endogenous auxin levels using the auxin biosynthesis inhibitor L-Kynurenine (L-Kyn) (*He et al., 2011*). Differentiation of chloronema to

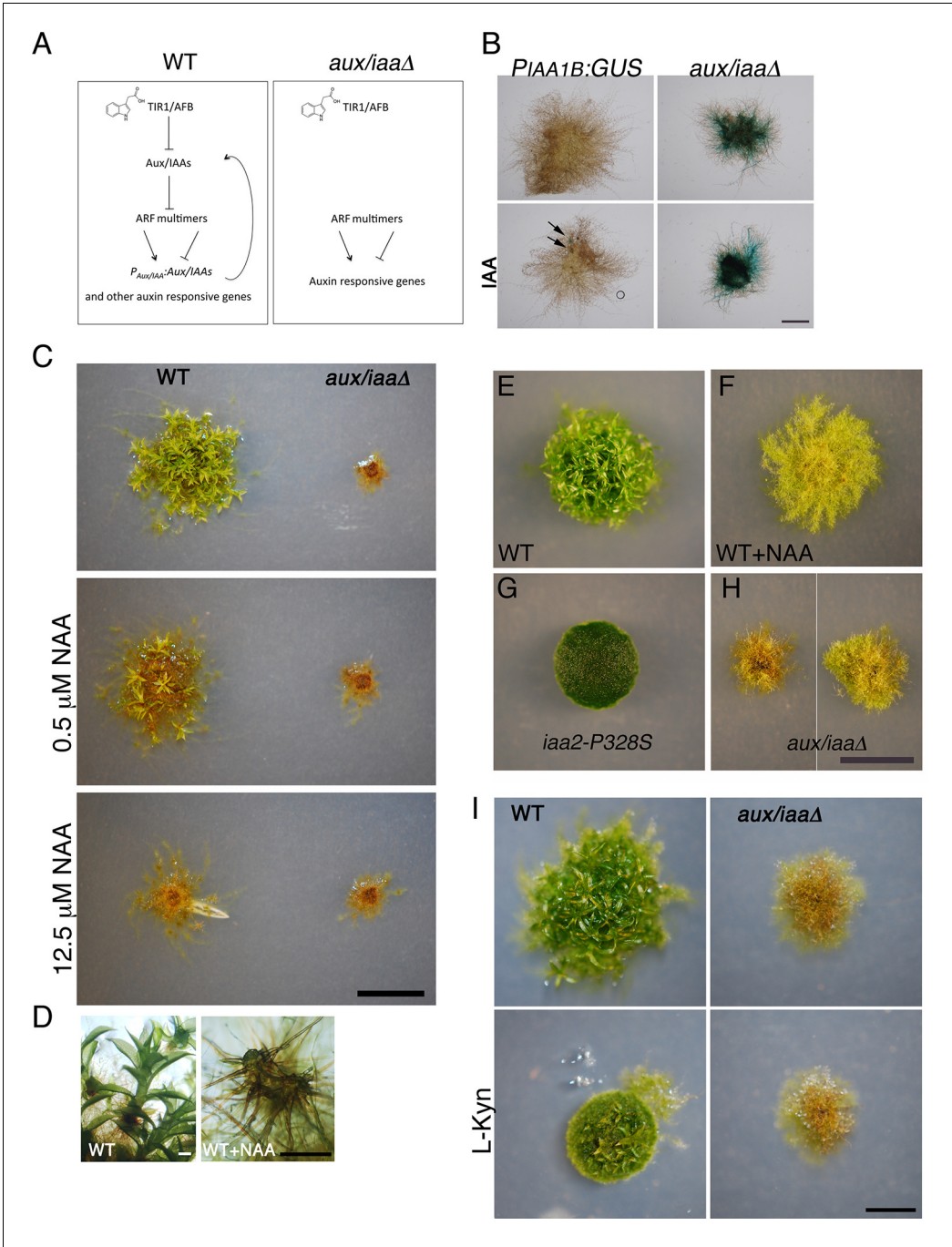

**Figure 1.** The *aux/iaaΔ* mutant displays a severe phenotype. (**A**) Scheme representing the auxin-signaling pathway in WT plants (left panel) and in the *aux/iaaΔ* mutant (right panel). (**B**) GUS expression in $P_{IAA1B}$:GUS and *aux/iaaΔ* plants carrying the $P_{IAA1B}$:GUS reporter. Arrows denote GUS expression. (**C**) WT and the *aux/iaaΔ* mutant grown for one month on BCD medium, stimulating gametophore development without auxin or with different concentrations of 1-naphthalene-acetic acid (NAA). (**D**) Microscopic enlargement of WT gametophores: left panel-leafy gametophores grown without exogenous auxin. Right panel-ectopic rhizoids emerging from a gametophore grown on 12.5 µM NAA. (**E–H**) Plants grown for one month on BCDAT medium to promote filamentous growth. (**E, F**) WT plant grown without auxin or with 12.5 µM NAA, respectively. (**G**) *iaa2-P328S* degron-motif mutant. (**H**) The *aux/iaaΔ* mutant has a variable phenotype. (**I**) WT and *aux/iaaΔ* mutant grown for one month on BCDAT or BCDAT supplemented with 10 µM L-Kyn. Scale bars: 1 mm (**B**), 0.5 cm (**C, H, I**), the scale bar in H also corresponds to **E–G**, 0.5 mm (**D**).

*Figure 1 continued on next page*

*Figure 1 continued*

The following figure supplements are available for figure 1:

**Figure supplement 1.** The *aux/iaaΔ* is a null mutant that displays constitutive auxin response.

**Figure supplement 2.** Early stages of filamentous growth of WT and *aux/iaaΔ* mutant.

---

caulonemal filaments and gametophore development in WT plants were slower in the presence of L-Kyn, whereas the *aux/iaaΔ* mutant was insensitive to the inhibitor (*Figure 1I*).

Next, we characterized the *aux/iaaΔ* mutant at earlier stages of growth following either protoplast recovery or tissue homogenization. Following protoplast recovery, WT plants had a higher growth rate compared to the mutant (*Figure 1—figure supplement 2A*). After seven days both chloronemal and caulonemal cells were observed in WT plants, whereas clearly differentiated cell types, either chloronemata or caulonemata, were not observed in the *aux/iaaΔ* mutant (*Figure 1—figure supplement 2B,C*). A comparison between WT chloronemal cells and the *aux/iaaΔ* mutant cells, seven days following tissue homogenization, revealed that the mutant cells are significantly wider (*Figure 2A,B*) suggesting that polarized cell growth is impaired. The chlorophyll concentration at that stage was lower in the mutant compared to WT (*Figure 2C*). Despite these significant morphological and physiological differences, the mutant filamentous growth appeared robust (*Figure 2—figure supplement 1*).

## The Aux/IAAs have a profound role in gene expression

The availability of the *aux/iaaΔ* mutant provided a unique opportunity to fully separate auxin perception from auxin response. Taking advantage of this opportunity, we profiled auxin-responsive transcription in WT and *aux/iaaΔ* protonemata by RNA sequencing (RNAseq) and compared the transcriptomes.

Our mutant analysis revealed that the differences between WT and the *aux/iaaΔ* mutant are more significant in later developmental stages. To minimize these differences, we performed RNAseq on protonemata grown on BCDAT for seven days after tissue homogenization (*Figure 2A*)

We first analyzed the WT auxin-responsive gene set following five hours of IAA treatment and identified 723 upregulated and 762 repressed genes [1.5 fold or more; p value adjusted (padj) <0.01] (*Supplementary file 1A–C*). Among the upregulated genes were several well-characterized early auxin response genes including the *Aux/IAA* genes as well as several genes homologous to *SAURs* and *LBD/ASLs* (*Paponov et al., 2008*). The *P. patens* genome encodes only two *GH3* homologs with a conserved function in IAA-conjugate synthesis (*Ludwig-Muller et al., 2009*). Neither of these genes displayed differential regulation in the auxin-responsive gene set and one of them was downregulated in the *aux/iaaΔ* mutant. A few *ARF* genes in Arabidopsis have been shown to respond to auxin (*Lau et al., 2011*; *Paponov et al., 2008*). Five out of sixteen *P. patens ARFs* (including both activating and repressing) were upregulated by auxin in the WT protonemata. Nine additional ARFs were upregulated in the *aux/iaaΔ* mutant compared to WT, indicating the presence of an ARF-dependent feedback loop that may have evolved in the ancestral land plant lineage. The larger number of auxin regulated ARFs in our datasets may reflect the fact that by comparing the *aux/iaaΔ* mutant to WT, we uncover all Aux/IAA-repressed genes, whereas the Arabidopsis transcriptome data describes the effect of exogenous auxin on a complex tissue and under specific conditions.

Strikingly, our results show that the *aux/iaaΔ* mutant is completely insensitive to auxin, with no changes in gene expression upon auxin treatment under our experimental conditions and statistical threshold (*Figure 2D*). A less stringent statistical threshold of padj <0.05 revealed only a few genes displaying a low fold change further validated this finding (*Supplementary file 1D*). These results imply that any factors that regulate auxin-dependent changes in gene expression must act through the Aux/IAAs, emphasizing the central role of these proteins in auxin signaling.

The *aux/iaaΔ* transcriptomic analysis revealed that a third of all annotated genes were differentially regulated in the mutant compared to WT (with 3752 and 4031 up- and downregulated genes, respectively), demonstrating the broad role of the TIR1-Aux/IAA pathway in land plant growth and

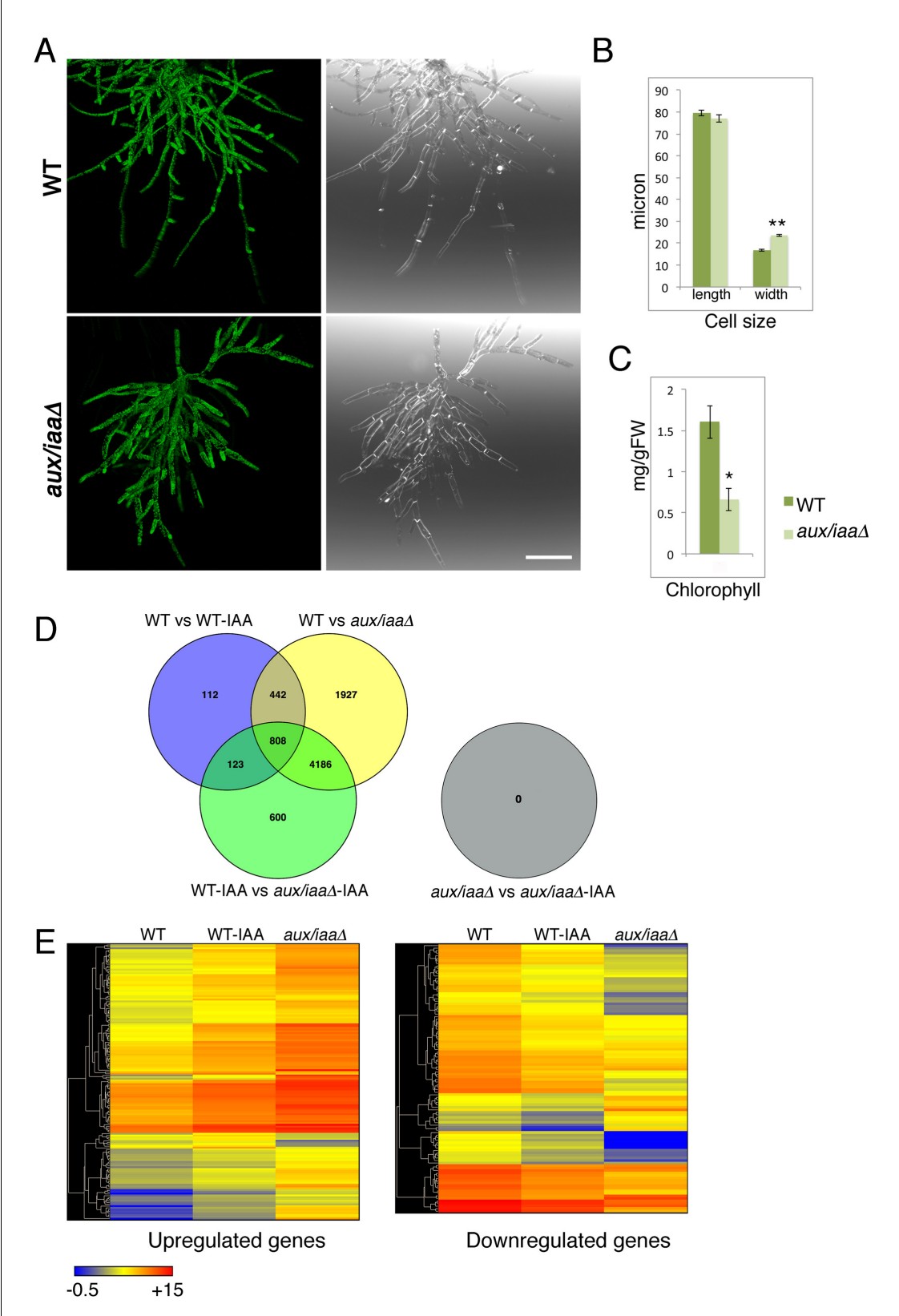

**Figure 2.** Loss of the *Aux/IAA* genes results in dramatic changes in gene expression at the filamentous developmental stage. (**A**) Confocal images of WT and *aux/iaaΔ* mutant protonemata seven days after tissue homogenization (Left panels-chloroplasts are visualized in green. Right panels-cell

*Figure 2 continued on next page*

*Figure 2 continued*

structures are visualized with DIC). Scale bar: 200 µm. (B) Average length and width (µm) of WT chloronemal cells and *aux/iaaΔ* mutant cells proximal to branch points. Error bars represent s.e.m. **p<0.001 (t-test), n=30. (C) Total chlorophyll concentration in WT and *aux/iaaΔ* protonemata seven days after tissue homogenization. Error bars represent s.e.m. *p<0.05 (t-test), n=3. (B) Venn diagram showing the overlap between the four data sets of differentially expressed genes (padj <0.01, fold change ≥1.5). (D) Hierarchical clustering of genes displaying differential expression between auxin treated and untreated WT plant samples with fold change ≥2 compared to their expression levels in the *aux/iaaΔ* mutant.

The following figure supplements are available for figure 2:

**Figure supplement 1.** Protonemal tissue grown under the growth conditions used for the RNAseq experiment.

**Figure supplement 2.** qPCR showing the expression levels of auxin responsive genes in WT plant and *aux/iaaΔ* mutant in mock- or 10 µM IAA-treatment for one hour.

**Figure supplement 3.** Graphical representation of enriched cellular components associated with auxin-responsive genes.

development. Strikingly, approximately 80% of these genes were not differentially regulated by auxin in WT plants whereas most of the regulated genes in WT were presented in the *aux/iaaΔ* mutant dataset (*Figure 2D*). Thus, our analysis allowed us to identify genes that were not revealed by auxin treatment. These may include genes that are regulated in a specific developmental, temporal, or environmental context as well as genes that are indirectly affected by the loss of the Aux/IAAs. Additionally, some genes may be revealed only when the Aux/IAAs are completely absent. For example, it is possible that very low levels of Aux/IAAs are sufficient to repress expression of some genes while for others the Aux/IAAs may be protected from auxin dependent-degradation. Finally, some genes may be represented by a small number of transcripts and/or exhibit a low level of differential expression, making them difficult to identify with standard procedures.

As illustrated by a hierarchical clustering of differentially expressed genes, the absence of the Aux/IAAs also had a strong effect on transcript levels (*Figure 2E*). Numerous up- and downregulated genes identified in WT plants displayed higher or lower expression levels in the *aux/iaaΔ* mutant respectively, indicating that many genes can be expressed over a broad range. For example, while the highest fold-change of differentially expressed genes following auxin treatment was fifteen, some of the same genes were differentially expressed in the *aux/iaaΔ* mutant with a fold-change of up to 430 compared to WT. These results indicate that significant levels of the Aux/IAA are present even after auxin treatment, further illustrating the robust nature of the auxin system. Several up- and downregulated genes that were previously described (*Lavy et al., 2012*), and genes displaying differential expression in this analysis, were selected as representative auxin-responsive gene markers, and used to validate the transcriptomic analysis (*Figure 2—figure supplement 2*).

The moss auxin-responsive transcriptome may facilitate discovery of ancestral auxin gene targets. We analyzed Gene Ontology (GO) terms associated with auxin responsive genes and found specific enriched categories for up- and downregulated genes. Analysis of upregulated genes revealed overrepresentation of genes involved in regulation of transcription and biosynthesis, whereas downregulated genes were highly enriched for processes occurring in the chloroplast and associated with photosynthesis, including light responses and carbon fixation (*Supplementary file 2A,B* and *Figure 2—figure supplement 3*). The set of downregulated genes differentially expressed in the mutant but not in WT presented a higher level of enrichment of these processes. Thus, our GO analysis suggests that in *P. patens* photosynthetic tissues, auxin signaling mediates separation of auxin-induced responses from energy production via induction of specific gene sets and the concurrent repression of others.

## The repressing ARFs regulate auxin-induced genes

The constitutive auxin response displayed by the *aux/iaaΔ* mutant and the broad effect of loss of Aux/IAA function on gene regulation highlight the central role of ARF transcription factors on gene expression. Based on sequence analyses and transient activity assays (*Tiwari et al., 2003*; *Ulmasov et al., 1999*), the ARF proteins were classified as either activators or repressors of transcription. While it is clear that the activating ARFs do activate transcription in plants and their mode

of action is beginning to emerge (*Wu et al., 2015*), this is not the case for the repressing ARFs. It has not been clearly demonstrated that the repressing ARFs act as transcriptional repressors in plants and a conceptual understanding of their role is lacking. For example, it is not known if repressing ARFs require the Aux/IAAs for their function or if the repressing and activating ARFs regulate the same genes. In addition it is not known if the repressing ARFs have a specific role in targeting downregulated genes.

The *aux/iaaΔ* mutant allows us to study the function of the repressing ARF in a relatively simple context lacking Aux/IAA repression and the negative feedback loops associated with their activity. Phylogenetic analyses reported that the *P. patens* ARF protein family consists of three clades (*Plavskin and Timmermans, 2012*) shared between all land plants and a fourth, non-seed-plant-specific clade characterized by the lack of the DNA binding domain [(*Paponov et al., 2009*), *Figure 3—figure supplement 1*]. One of the clades, comprising four proteins, groups with Arabidopsis repressing ARFs, whereas another clade, comprising seven ARF proteins, groups with Arabidopsis activating ARFs. The *M. polymorpha* ARF proteins were recently shown to affect transcription in a transient expression assay in accordance with their phylogenetic classification (*Kato et al., 2015*) strongly suggesting that ARF function is conserved in land plants. Furthermore, in *P. patens*, higher levels of putative repressing ARFs resulted in decreased auxin response (*Plavskin et al., 2016*). We selected *ARFb4* as a representative repressing ARF, and overexpressed an ARFb4 c-Myc fusion in the *aux/iaaΔ* mutant background (*ARFb4OE_aux/iaaΔ*). We found that overexpression of ARFb4 suppressed the constitutive auxin phenotype of the *aux/iaaΔ* mutant including the formation of green chloronemal-like filaments (*Figure 3A*). In fact, in a high expressing transgenic line (*Figure 3A* #1 and *Figure 3—figure supplement 2C*), the phenotype was quite similar to that of dominant auxin-resistant *aux/iaa* mutants [*Figure 1G* and (*Prigge et al., 2010*)]. This result demonstrates that repressing ARFs do function as repressors in plants and that this activity does not require the Aux/IAAs. It is worth noting that ARFb4, as well ARFb2 and ARFb3 do not contain any of the known motifs that are thought to recruit TPL and therefore may not affect chromatin remodeling directly. To define the mechanism underlying the suppression of the *aux/iaaΔ* phenotype we assessed the expression of the $P_{IAA1B}$:*GUS* marker and other auxin-responsive genes. The GUS marker (*Figure 3B*), as well as the additional reporter genes (*Figure 3C*) had dramatically lower expression levels compared to the *aux/iaaΔ* mutant indicating that activating ARFs and the repressing ARFs can affect, either directly or indirectly the same target genes. Further, the auxin response genes did not respond to auxin in the *ARFb4OE_aux/iaaΔ* line confirming that overexpression of the repressing ARF converted the constitutive auxin response line to an auxin resistant line. Overexpression of Arabidopsis ARF1 resulted in similar effects, indicating that the role of repressing ARFs is conserved between *P. patens* and Arabidopsis (*Figure 3C* and *Figure 3—figure supplement 2*).

Although ARFb4 overexpression resulted in a clear suppression of the *aux/iaaΔ* constitutive response, the phenotype was not as dramatic as the *aux/iaa* degron mutants. Expression of the auxin reporter genes was lower in the *iaa2-P328S* mutant compared to the *ARFb4OE_aux/iaaΔ* (*Figure 3C* #5). This result suggests that the repression conferred by the Aux/IAAs is stronger than ARF-based repression.

Our results clearly indicate that auxin cannot stimulate transcriptional response in the *ARFb4OE_aux/iaaΔ* following several hours of treatment. However, it is possible that auxin can promote protonemal development through other mechanisms or by longer treatment. To explore this possibility we grew the *ARFb4OE_aux/iaaΔ* #1 line on NAA. Filament differentiation was not observed after one month of auxin treatment (*Figure 3D*) indicating that the Aux/IAA function is required for developmental transition in protonema.

We next examined how repressing and activating ARFs interact to regulate the same gene targets. Several attempts to overexpress the activating ARFs *P. patens* ARFa8 and Arabidopsis ARF7 in the *aux/iaaΔ* mutant were unsuccessful. We only recovered one *ARFa8* transgenic line with weak expression of ARFa8 suggesting that further gene activation, beyond that observed in the *aux/iaaΔ* mutant, is lethal. As an alternative approach we selected one *ARFb4OE_aux/iaaΔ* line (*Figure 3A* #1) and introduced an inducible *ARFa8-glucocorticoid receptor (GR)* fusion gene into this line (*ARFa8-GR_ARFb4OE_aux/iaaΔ*). Two resulting lines were selected for further analysis (*ARFa8-GR_ARFb4OE_aux/iaaΔ* #1 & #2) (*Figure 4A–C*). Following dexamethasone (DEX) treatment these lines developed gametophores with ectopic rhizoids indicative of auxin hypersensitivity (*Figure 4A,B*). Expression of the auxin reporter genes supported the phenotypic analysis since expression levels of

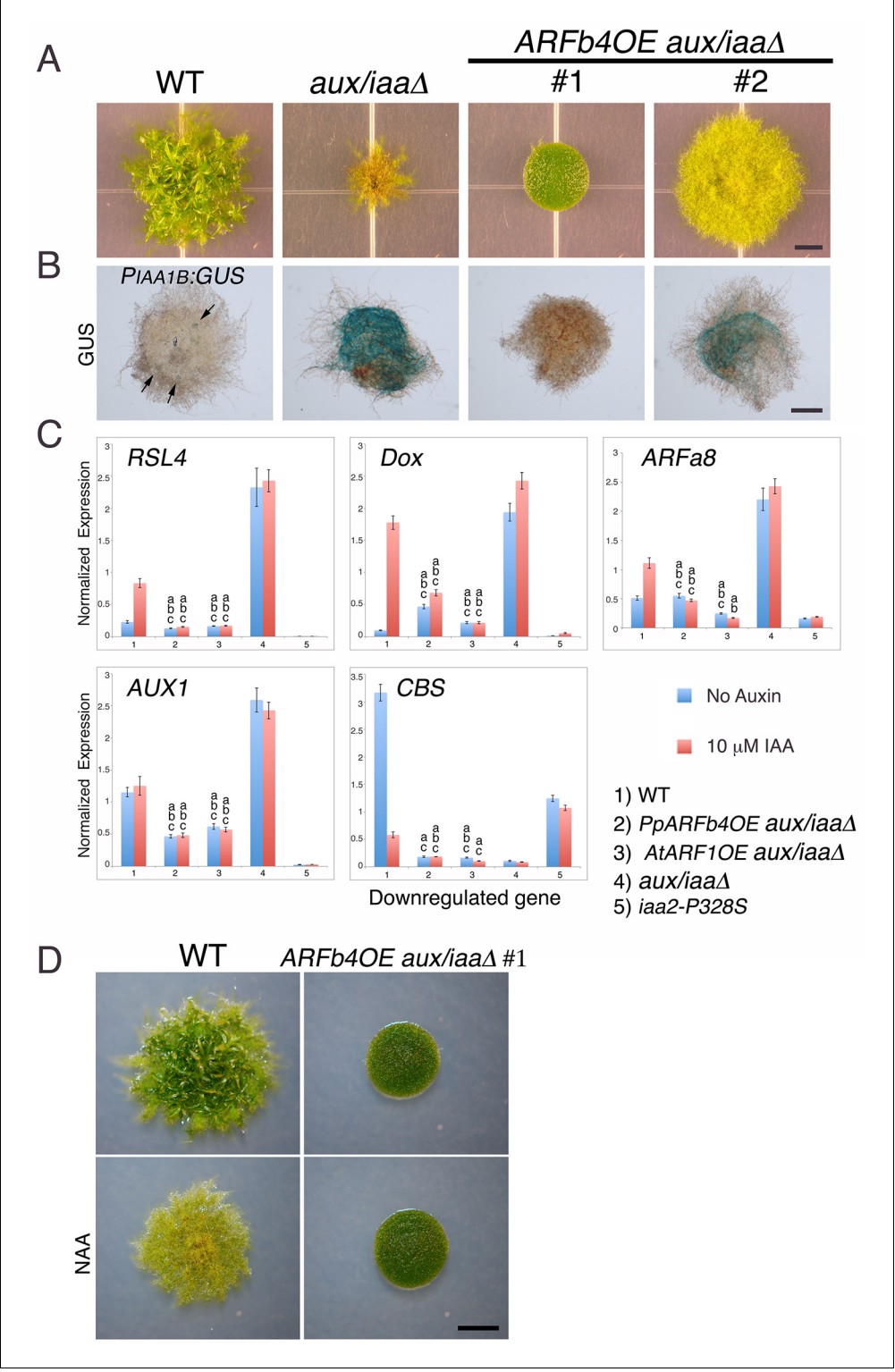

**Figure 3.** Repressing ARFs target auxin-induced genes. (**A**) WT, *aux/iaaΔ* and two *aux/iaaΔ* lines overexpressing ARFb4 (*ARFb4OE_aux/iaaΔ*) grown for one month on BCDAT. (**B**) GUS expression in *aux/iaaΔ* and two *ARFb4OE_aux/iaaΔ* lines carrying the P*IAA1B*:*GUS* reporter. Arrows denote GUS expression. (**C**) qPCR showing the expression levels of auxin responsive genes in WT, *ARFb4OE_aux/iaaΔ* (line #1), *AtARF1_aux/iaaΔ*, *aux/iaaΔ*, and *iaa2-P328S* in presence of 10 µM IAA-treated for five hours or mock. Error bars represent s.e.m. a/b/c=P<0.05 (t-test), n=3. a=t-test comparing the lines to WT, b= t-test comparing the lines to *aux/iaaΔ*. c=t-test comparing the

*Figure 3 continued on next page*

*Figure 3 continued*
lines to *iaa2-P328S*. (D) WT and *ARFb4OE_aux/iaaΔ* #1 grown on BCDAT without auxin or with 12.5 μM NAA for
one month. Scale bars: 0.5 cm (A), 0.5 mm (B), 0.5 cm (D)
The following figure supplements are available for figure 3:
**Figure supplement 1.** Phylogeny of Land Plant ARFs.
**Figure supplement 2.** The Arabidopsis repressing ARF1 has similar effects on plant growth as ARFb4.

the upregulated genes in the DEX treated plants were higher compared to both the *ARFb4OE_aux/iaaΔ* background and untreated *ARFa8-GR_ARFb4OE_aux/iaaΔ* transgenic lines (*Figure 4D*). Similarly, the level of a representative downregulated gene was lower (*Figure 4D-CBS*). This demonstrates that ARFb4 and ARFa8, as representatives of repressing and activating factors respectively, display opposing activity on the same gene targets.

Activating and repressing ARFs may target the same genes by competing for the same promoter elements, by binding to different regions within the same promoters, or through an indirect mechanism. To distinguish between these possibilities we performed an electrophoretic mobility shift assay with DNA sequences from the *DR5* reporter and representative auxin responsive gene promoters, *IAA1A, IAA1B*, and *ARFb4*. The results indicated that ARFb4 and ARFa8 DNA binding domains can bind to the same DNA sequences (*Figure 4E*).

## The ARFs and the Aux/IAAs display complex interactions

Our results demonstrate that both the repressing ARFs and Aux/IAAs are capable of gene repression, albeit to different extents. To learn more about these two types of repression, we determined the effects of loss of repressing ARFs either in the presence or absence of the *Aux/IAAs*. The coding regions of *ARFb2* and *ARFb4* were deleted in WT as well as in the *iaa1b iaa2Δ* double and *aux/iaaΔ* triple mutants (*Figure 5—figure supplement 1*). The loss of *ARFb2* and *ARFb4* in the *aux/iaaΔ* line (*arfb2 arfb4Δ aux/iaaΔ*) did not result in any morphological changes or a clear trend with respect to changes in gene expression (*Figure 5—figure supplement 2*). Conversely and surprisingly, the *arfb2 arfb4* knockout in both the WT and the *iaa1b iaa2Δ* double mutant (*arfb2 arfb4Δ and arfb2 arfb4Δ iaa1b iaa2Δ*, respectively) resulted in lower expression levels of some of the upregulated auxin responsive genes, and higher levels of downregulated genes compared to their backgrounds (*Figure 5A*). Although the expression of these genes in both untreated and auxin-treated *arfb2 arfb4Δ* plants was reduced, the fold change was either similar or even higher compared to the WT and *iaa1b iaa2Δ* backgrounds (*Figure 5A*, *RSL4*: 7–3 fold higher and *Dox*: 2–1.4, respectively). Phenotypic analysis of the *arfb2 arfb4Δ* revealed developmental defects including shorter filaments and fewer leafy gametophores consistent with reduced auxin response (*Figure 5—figure supplement 3*). These findings further support our hypothesis that repressing ARFs can affect the same genes as activating ARFs. However, they revealed an unexpected trend in which the loss of repressing ARFs affects transcription in the same direction as their overexpression. To further confirm these contradictory results we also used transient RNAi (*Bezanilla et al., 2005*) targeting the four repressing ARFs (*arfb1-4*). The RNAi construct was expressed in a transgenic plant expressing the synthetic auxin-responsive reporter *DR5:DsRED*. In agreement with the analysis of the *arfb2 arfb4Δ* stable lines the resulting transformants displayed a lower expression level of the *DR5:DsRED* reporter compared to the control plants when treated with auxin, indicative of reduced auxin sensitivity of the *arfb1-4* RNAi knockdown lines (*Figure 5B*).

These contradictory results may be explained by the indirect effect of negative feedback loops, in which the extensive production of Aux/IAA proteins could result in an overall decrease in auxin response. However, this possibility does not appear to explain the phenotype of *arfb2 arfb4Δ* lines, as the transcript levels of *IAA1A* (*Figure 5A-IAA1A*) are low. Alternatively, it is possible that both activating and repressing ARFs as well as the Aux/IAAs jointly coordinate gene induction. This hypothesis is consistent with our analysis of the *arfb2 arfb4Δ* lines. In the presence of the Aux/IAAs, loss of ARFB2 and ARFB4 results in reduced auxin response. However, this effect is suppressed by

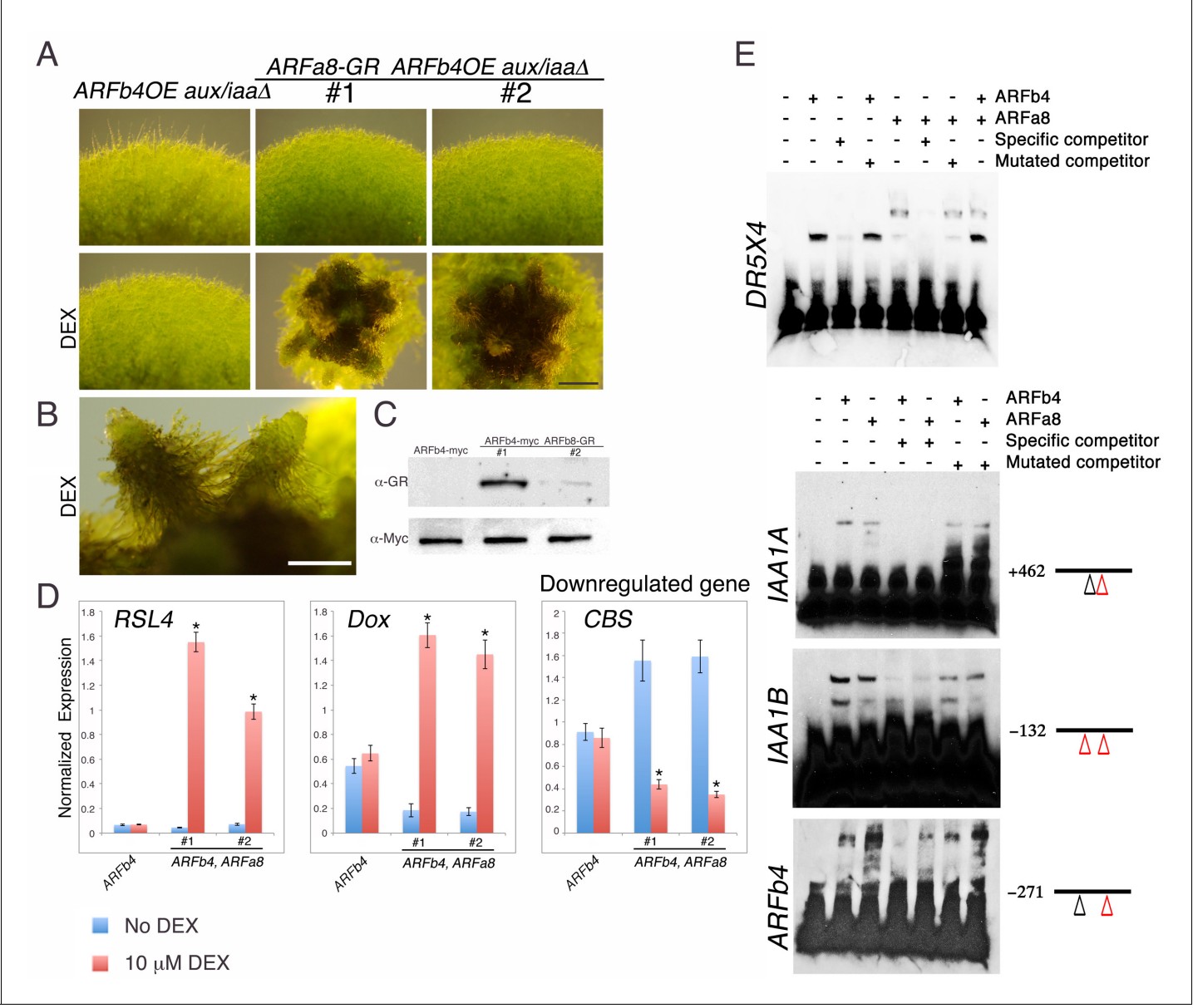

**Figure 4.** Repressing and activating ARFs display opposite effects on the same target genes. (**A**) *ARFb4OE_aux/iaaΔ* (line #1 in *Figure 3A*), and two *ARFa8-GR ARFb4OE_aux/iaaΔ* lines grown on BCDAT or BCDAT supplemented with 10 μM DEX for a month. Scale bar: 0.5 mm. (**B**) Enlargement of gametophores of line #1. Scale bar: 0.5 mm. (**C**) Immunoblot of total protein extracts from plants shown in E, detected with GR and c-Myc antibodies. (**D**) qPCR showing the expression levels of auxin responsive genes in the transgenic lines shown in A, *ARFb4OE_aux/iaaΔ* (*ARFb4*) and *ARFa8-GR ARFb4OE_aux/iaaΔ* (*ARFb4, ARFa8*) mock- or 10 μM DEX-treated for overnight. Error bars represent s.e.m. *p<0.05 (t-test comparing DEX-induced- to uninduced gene expression levels in *ARFb4, ARFa8* lines), n=3. (**E**) Electrophoretic mobility shift assay with GST-ARFb4 DBD and GST-ARFa8 DBD (ARFb4, ARFa8) with biotin-labeled DNA probes from *DR5*, *IAA1A*, *IAA1B*, and *ARFb4* promoters. + and – signs denote the presence or the absence of ARF proteins, and unlabeled specific or mutant competitor DNA sequences. Black line and numbers represent the probe locations respectively to the gene transcription start sites. Red triangles: canonical TGTCTC AuxREs. Black triangles: core TGTC AuxREs.

the deletion of the *Aux/IAA* genes in the *arfb2 arfb4Δ aux/iaaΔ* line (*Figure 5—figure supplement 2* compared to *Figure 5A*). Since the activating and repressing ARFs can target the same genes, loss of the repressing ARFs may result in increased levels of activating ARFs on the auxin-responsive promoters. This may have the unexpected effect of recruiting more Aux/IAA repressors to these promoters. To test this idea we overexpressed the activating ARFa8, in WT plants (*ARFa8OE*). The

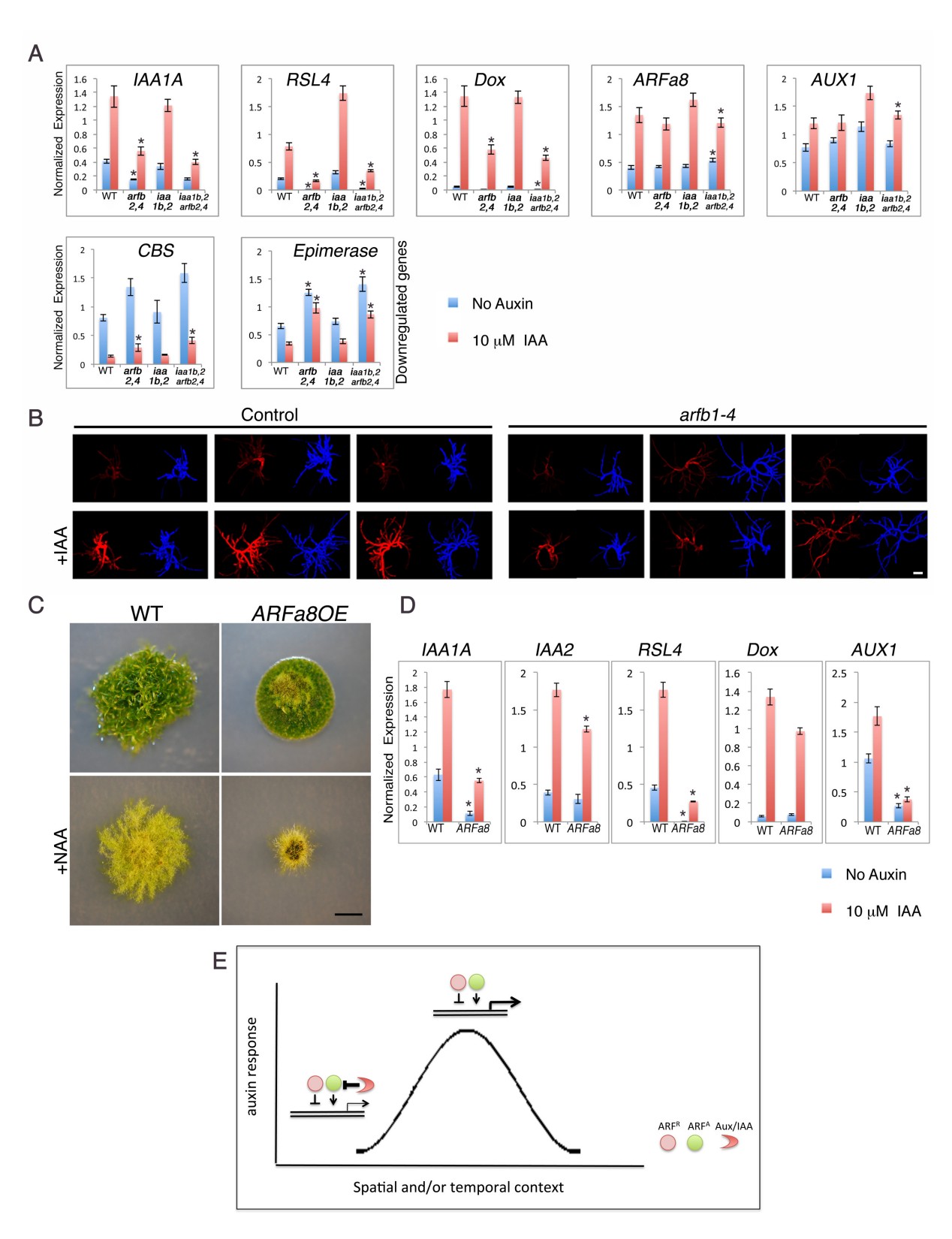

**Figure 5.** Both repressing and activating ARFs affect auxin response in the same direction in the presence of the *Aux/IAAs*. (**A**) qPCR showing the expression levels of auxin responsive genes in WT, *arfb2 arfb4Δ (arfb2,4)*, *iaa1b iaa2Δ (iaa1b,2)*, and *arfb2 arfb4 iaa1b iaa2Δ (iaa1b,2 arfb2,4)*, mock- or

*Figure 5 continued*

10 µM IAA-treated for five hours. Error bars represent s.e.m. *p<0.05 (t-test comparing the *arfb2 arfb4*Δ, and *arfb2 arfb4 iaa1b iaa2*Δ lines to either WT or *iaa1b iaa2*Δ, respectively), n=3. (**B**) Three independent *arfb1-4* RNAi and control lines mock- or 10 µM IAA-treated. Each line is presented by two images (DsRED fluorescence (red), chlorophyll auto-florescence (blue). Each image is from projection stacks of multiple confocal sections. Scale bar: 100 µm. (**C**) WT and an ARFa8 overexpression line (*ARFa8OE*) grown for one month on BCDAT without auxin or with 12.5 µM NAA. Scale bar: 0.5 cm. (**D**) qPCR showing the expression levels of auxin responsive genes in WT and *ARFa8OE* mock- or 10 µM IAA-treated for five hours. Error bars represent s. e.m. *p<0.05 (t-test comparing the *ARFa8OE* line to WT), n=3. (**E**) Model for auxin regulation of transcription. The expression level of an auxin responsive gene is determined by the interplay between the Aux/IAAs, the repressing ARFs, and the activating ARFs. At low auxin levels the Aux/IAAs provide stringent repression. At high auxin levels, the Aux/IAAs are degraded resulting in increased transcription through the action of the activating ARFs. The repressing ARFs act to buffer gene expression by attenuating the activity of the activating ARFs. The interplay between the three protein groups results in a wide range of gene expression levels that contribute to a dynamic and context specific auxin response.

The following figure supplements are available for figure 5:

**Figure supplement 1.** PCR detecting the insertion position of different *arfb2 arfb4*Δ mutants.

**Figure supplement 2.** *Arfb2* and *arfb4* Knockout in *aux/iaa*Δ mutant background (*arfb2 arfb4*Δ *aux/iaa*Δ) does not result in phenotypes comparable to reduced auxin sensitivity.

**Figure supplement 3.** Phenotypic analysis of *arfb2 arfb4*Δ in WT and *iaa1b iaa2*Δ mutant background.

resulting lines displayed a phenotype characteristic of auxin-resistant plants and exhibited reduced expression of auxin responsive genes (*Figure 5C,D*), which supports our hypothesis. Collectively, our results support a model in which repressing ARFs can affect the occupancy of activating ARFs and Aux/IAAs on auxin responsive promoters. These interactions may provide buffering capacity that allows fine-scale regulation of auxin responsive genes (*Figure 5E*).

## Discussion

In this study we exploited the relative simplicity of *P. patens* to address the complexity of auxin-regulated transcription. The Aux/IAA proteins have a central role in auxin signaling, serving as both auxin co-receptors and transcriptional repressors. However, genetic analysis of these genes has been limited in flowering plants because of genetic redundancy. By deleting all three *Aux/IAAs* in moss we have determined the effects of the complete absence of Aux/IAA function on plant growth and auxin response. Remarkably, we found that the Aux/IAAs have a broad impact on the expression levels of a great number of genes, the majority of which do not respond to auxin treatment of moss protonemata. Thus our analysis demonstrates that existing transcriptomic studies conducted in flowering plants may significantly underestimate the effects of auxin on the genome and illustrates the central role of the TIR-Aux/IAA pathway on plant growth and development. While our work highlights the remarkably broad impact of the Aux/IAA proteins on gene expression, it also demonstrates an absolute requirement for the Aux/IAAs in auxin-responsive transcription. Some auxin-mediated signaling pathways have been proposed to affect transcription independently of the TIR1/AFB-Aux/IAA signaling cascade. These include a negative role of Mitogen-Activating Protein Kinase (MAPK) activity on auxin-regulated genes (*Lee et al., 2009*), and the effect of the F-box protein SKP2A on the stability of cell division transcription factors (*Jurado et al., 2010*). Our findings indicate that at least in moss, any possible effect of these pathways on auxin-regulated transcription requires Aux/IAAs' function.

In *P. patens*, auxin-dependent modulation of gene expression triggers developmental transitions and differentiation. Our mutant analyses reveal that the transition from chloronema to caulonema is regulated by a dynamic balance of quantitative auxin responses. When auxin response is low, as in the auxin-resistant mutants, protonema growth is not accompanied by developmental transition to the caulonemal stage. In contrast, a constitutive auxin response, as exhibited by the *aux/iaa*Δ mutant, results in rapid and abnormal maturation.

It has been proposed that repressing and activating ARFs might compete for binding to the same promoters (*Vernoux et al., 2011*). Structural studies of Arabidopsis ARF1 and ARF5 revealed that both classes of ARF can potentially bind to similar cis elements and form high order oligomers of

ARFs and Aux/IAAs (*Boer et al., 2014*; *Korasick et al., 2014*). Our work provides experimental evidence that auxin responsive genes are targeted by both repressing and activating ARFs, which in turn coordinate their levels of expression. The structural studies as well as our results can support two models of ARF activity. Repressing and activating ARFs can either compete on the same binding sites or cooperatively induce transcription by forming heterodimers. Despite its classification as an activating ARF, the Arabidopsis ARF5 was shown to repress some of its identified gene targets (*Zhang et al., 2014*; *Zhao et al., 2010*). Given that ARF transcription factors may display opposing roles in different contexts, ARF activity can result in a very complex gene expression pattern. These mechanisms, by which different ARFs coordinate transcriptional responses, can explain the broad effect of auxin on gene expression and the complexity of the resulting transcriptional networks as each gene can display a wide range of expression levels. Recent studies in liverwort support this hypothesis. The *M. polymorpha* encodes only one Aux/IAA and one member of each ancient ARF clade yet these transcription factors are sufficient to pattern a complete body plan (*Flores-Sandoval et al., 2015*; *Kato et al., 2015*).

Unlike the activating ARFs, our understanding of the repressing ARFs is limited. For some repressing ARFs, the presence of an EAR domain suggests that repression could involve recruitment of the TPL co-repressor. However, many repressing ARFs, including PpARFb2, PpARFb4, and AtARF1 used in this work do not have this motif. We show that repression provided by these repressing ARFs is weaker compared to the repressing effect of the Aux/IAAs. Based on this observation, we suggest that repressing ARFs may fine-tune gene expression. The Aux/IAAs repress gene expression through interactions with co-repressors such as TPL and the subsequent recruitment of chromatin remodeling factors (*Kagale and Rozwadowski, 2011*). This may provide stable and long term repression when auxin levels are low. In contrast, when auxin levels are high and/or when auxin response needs to be dynamic, the repressing ARFs may provide a less stable repression that fine-tunes auxin response in the absence of the Aux/IAAs. The fact that the repressing *ARF* genes are induced by auxin and therefore constitute a negative feedback module is consistent with this idea.

In flowering plants, auxin integrates light signals and growth responses (*Halliday et al., 2009*). While we have some knowledge of how light affects auxin synthesis and distribution, the effects on auxin transcriptional responses are less known. In moss, the interplay between light, auxin, and growth has been implicated in the transition from chlronema to caulonema. In addition to auxin, light intensity and the availability of nutrients affect the transition from one cell type to the other. High light and glucose triggers caulonema formation while low light inhibits caulonemal growth and stimulates chloronemal branching (*Thelander et al., 2005*). These observations suggest that favorable conditions promote the differentiation of elongated caulonemal cells, whereas low energy conditions promote the formation of chloroplast-rich chloronemal cells thus increasing photosynthetic capacity and energy production. We found that the auxin downregulated gene set is dominated by genes involved in photoperception and carbon fixation. This finding establishes the molecular basis for auxin signaling in response to light stimuli and suggests that auxin may function as a molecular switch between energy production and growth.

Our model cannot explain a direct effect of the ARF transcription factors on downregulated genes. Only a few repressed genes have been identified after a short auxin treatment (*Chapman et al., 2012*; *Paponov et al., 2008*). Based on the GO analysis presented here it seems likely that transcription factors that are directly induced by auxin repress many processes downstream of auxin, particularly those associated with photosynthesis. These auxin-induced regulators and their downstream targets are yet to be defined.

## Materials and methods

### Moss strains and growth conditions

WT 'Gransden-2004' and mutant *P. patens* strains were grown at 25°C under continuous light at an intensity of 40–70 µmol/m2/s on BCD or BCDAT (BCD supplemented with 5 mM Ammonium Tartrate) media. Growth and differentiation of protonema is slower on BCDAT compared to BCD. For each experiment, a medium was selected to allow for analysis of either protonema development or filamentous and gametophyte differentiation, respectively.

## Molecular cloning of gene disruption and expression constructs

PCR-amplified DNA fragments were cloned into pENTR-D/TOPO (Life Technologies), and were subsequently cloned either as digested fragments or by LR-Clonase-mediated recombination (Life Technologies) into the final vectors as detailed in the following procedures (Primer pairs used for amplification are listed in *Supplementary file 3*. backbone vectors are listed in *Supplementary file 4*). *Gene knockout constructs:* Approximately one kilobase of genomic sequence upstream (5' region) and downstream (3' region) to the genes coding region were amplified and cloned into either pBNRF, pBNRF-GUS, or pBHRF2 as specified bellow:

| Genes | | Primers | Cloning procedure | Backbone |
|---|---|---|---|---|
| *IAA1A* | 5' region | PML61, 62 | *Bam*HI ligation | pBHRF2 |
| | 3' region | PML59, 60 | *Spe*I ligation | |
| *IAA1B* | 5' region | PML52, 53 | *Bam*HI ligation | pBNRF-GUS |
| | 3' region | PML59, 60 | *Spe*I ligation | |
| *IAA2* | 5' region | PML63, 64 | *Bam*HI ligation | pBNRF |
| | 3' region | PML65, 66 | *Spe*I ligation | |
| *ARFb4* | 5' region | PML599, 600 | *Bam*HI ligation | pBNRF-GUS/ pBNRF |
| | 3' region | PML601, 602 | *Spe*I ligation | |
| *ARFb2* | 5' region | PML677, 678 | *Bam*HI ligation | pBHRF2 |
| | 3' region | PML679, 680 | *Sal*I ligation | |

*Gene replacement construct:* To create a mutant degron motif of IAA2 (*iaa2-P328S*) the *PpIAA2* genomic region was amplified from the *ppiaa2-183* mutant (*Prigge et al., 2010*) using the *PpIAA2* genomic region-F' and R' primer pair. The fragment was subcloned as an *Avr*II and *Bam* HI fragment. *Overexpression and inducible expression constructs:* Constructs for protein-expression were generated by LR-Clonase-mediated recombination between pENTR-D/TOPO plasmids and destination vectors. To create *PpARFb4*, *PpARFa8*, and *AtARF1 c-myc* fusions, the genes' coding regions were amplified using the primer pairs PML455 & 457, PML461 & 463, and PML776 & 777 respectively and recombined into pMP1377. To create *ARFa8* and glucocorticoid receptor fusion (*ARFa8-GR*), the rat glucocorticoid receptor DNA fragment was inserted into the *Asc*I site of *ARFa8*- pENTR-D/TOPO plasmid to create an *ARFa8-GR* fusion. The resulting recombinant fusion was recombined into pTHUBiGate.

To create a construct carrying the auxin responsive marker (*DR5:DsRED2*) eight repeats containing the DR5rev element (ggGAGACAttt) were inserted ahead of a minimal *CaMV 35S* promoter (-50 to +26). This promoter fragment was fused to the *DsRED2* coding region by PCR then inserted together as a *Bam*HI fragment into pMP1432 (replacing the *Hsp*:Gateway cassette).

## Moss transformation and screening of transgenic lines

Protoplast isolation and PEG-mediated transformation of *P. patens* was performed as described in (*Nishiyama et al., 2000*). Three to five days after regeneration, transformants were selected on BCDAT medium containing 20 mg/l of either G418 or hygromycin, or 150 mg/l Gentamycin. Plants transformed with *iaa2-P328S* fragment were identified based on their phenotype on media containing 20 µM NAA.

Transgenic knockout lines were screened by PCR for the presence of both left and right transgene-endogenous sequence junctions to verify the insertion of the transgene to the targeted locus and for the absence of the corresponding coding region. cDNA was detected by RT-PCR to confirm the absence of a transcribed product. Genomic DNA extracted from over- and inducible expression transgenic lines was detected by PCR to confirm the presence of the transgene, and recombinant protein expression was detected by immunoblot as described in Supplemental Experimental Procedures. *iaa2-P328S* lines were genotyped by derived Cleaved Amplified Polymorphic sequences (dCAPS) (Table S7).

All knockout mutants expressed either *nptII* or *Hygromycin* selectable marker genes. To create higher order mutants, the selectable marker cassettes were first excised using the site-specific recombination Cre/lox system (*Schaefer and Zryd, 2001*) to allow recycling of the selectable marker genes. The resulting lines are listed in the following table.

## Strains used in this study

| Description | Name in the text | Source |
|---|---|---|
| $P_{IAA1B}$:GUS/iaa1b$\Delta$ | $P_{IAA1B}$:GUS | This work |
| $P_{IAA1B}$:GUS/iaa1b$\Delta$   iaa2$\Delta$ | iaa1b iaa2$\Delta$ | This work |
| $P_{IAA1B}$:GUS/iaa1b$\Delta$   iaa2$\Delta$   iaa1a$\Delta$ | aux/iaa$\Delta$ | This work |
| $P_{ARFb4}$:GUS/arfb4$\Delta$ | | This work |
| $P_{ARFb4}$:GUS/arfb2$\Delta$   arfb4$\Delta$ | Arfb2 arfb4$\Delta$ | This work |
| arfb2$\Delta$ | | This work |
| $P_{IAA1B}$:GUS/iaa1b$\Delta$ iaa2$\Delta$ arfb2$\Delta$ | | This work |
| $P_{IAA1B}$:GUS/iaa1b$\Delta$   iaa2$\Delta$   arfb2$\Delta$ arfb4$\Delta$ | arfb2 arfb4$\Delta$ iaa1b iaa2 | This work |
| $P_{IAA1B}$:GUS/iaa1b$\Delta$   iaa2$\Delta$   iaa1a$\Delta$   arfb2$\Delta$ arfb4$\Delta$ | arf2b arf4b aux/iaa$\Delta$ | This work |
| $P_{IAA2}$:iaa2-P328S G4 GH3:GUS | iaa2-P328S | This work |
| $P_{UBi}$:PpARFa8-c-myc | ARFa8OE | This work |
| $P_{IAA1B}$:GUS/iaa1b$\Delta$   iaa2$\Delta$   iaa1a$\Delta$   $P_{UBi}$:PpARFb4-c-myc | ARFb4OE_aux/iaa$\Delta$ | This work |
| $P_{IAA1B}$:GUS/iaa1b$\Delta$   iaa2$\Delta$   iaa1a$\Delta$   $P_{UBi}$:AtARF1-c-myc | AtARF1OE_aux/iaa$\Delta$ | This work |
| $P_{IAA1B}$:GUS/iaa1b$\Delta$   iaa2$\Delta$   iaa1a$\Delta$   $P_{UBi}$:PpARFb4-c-myc $P_{UBi}$:PpARFa8-GR | ARFa8-GR_ARFb4OE_aux/iaa$\Delta$ | This work |
| NLS4 | | (*Bezanilla et al., 2005*) |
| DR5:dsRED NLS4 | DR5:dsRED | This work |
| G4 GH3:GUS | | (*Bierfreund et al., 2003*) |

## RNA-sequencing

Protonemal tissue comprised mainly of chloronemata from WT and *aux/iaa* knockout mutant plants was homogenized in a Waring blender and plated in triplicate on BCDAT plates with cellophane overlays for seven days. The cellophane overlays covered with protonemal tissue were transferred into liquid BCD medium containing either 10 µM IAA or the equivalent amount of ethanol solvent and incubated at 25°C under continuous light for 5 hr. Total RNA was isolated using the RNeasy plant mini kit (Qiagen) and treated with DNA-free™ DNAse removal kit (Life technologies). The three biological replicates were sequenced on Illumina HiSeq2500 by New York Genome Center, resulting in 30 million 50 bp paired-end reads per sample. Total reads were mapped to the *P. patens* genome (version 1.2.1) using RNAseq aligner STAR (*Dobin et al., 2013*). Genes were quantified with featureCounts (*Liao et al., 2014*), using v6 annotation the GTF file corresponding to the annotation v6 (obtained from http://phytozome.jgi.doe.gov/). Differential Expression was determined using the DESeq2 (*Love et al., 2014*). The sequences reads have been deposited to the sequence Read Archive (SRA) database (BioProject accession PRJNA317343).

## RNA-sequencing, additional analysis tools

GO enrichment analysis was carried out using AgriGO (*Du et al., 2010*) with GO annotation obtained from https://www.cosmoss.org. Hierarchical clustering was performed with DNASTAR.

## RNA interference

RNA interference was carried out as described in *Bezanilla et al. (2005)*. Overlapping PCR fragments from *PpARFb4* (PML506, 507) and *PpARFb3* (PML508, 509) were fused by PCR. Overlapping PCR fragments from *PpARFb1* (PML510, 511) and *PpARFb2* (PML512, 513) were fused by PCR. The resulting two fragments were fused together by PCR and cloned into pENTR-D/TOPO. The hairpin

expression plasmids were generated by LR-Clonase-mediated recombination between the resulting pENTR plasmid and pUGGi. A pUGi vector lacking the gateway cassettes was used as control. The hairpin expression plasmids were transformed into *DR5:DsRED* strain which expresses a nuclear-localized GFP-GUS fusion protein. Following protoplast regeneration, the transformants, carried on cellophane overlays, were transferred to BCD media with Hygromycin and grown for eight days and then transferred into liquid BCD medium containing either 10 μM IAA or ethanol and incubated for 32 hr. Transformants lacking GFP fluorescence were selected and photographed. Fluorescence signals were detected for GFP (excitation 488 nm, emission 493–535 nm), DsRED (561 nm excitation, 566–623 nm emission), and chlorophyll auto-fluorescence (excitation 633 nm, emission 703–735 nm), using laser scanning confocal microscope (Zeiss MLS 710).

## qRT-PCR

Protonemal tissue was grown in triplicate or quadruplicates on BCDAT plates with cellophane overlays for seven days. For IAA or DEX treatment, plant tissue was transferred into liquid BCD medium containing either 10 μM IAA or 10 μM DEX or the equivalent amount of ethanol. Following incubation, the tissue was collected and total RNA was isolated using RNeasy plant mini kit (Qiagen). 500 μg RNA was reverse transcribed using the Superscript III First Strand cDNA Synthesis System (Life Technologies). 20 μl RT reaction was diluted with water to a final volume of 200 μL. PCR samples contained 4 μl diluted cDNA were detected using the CFX Connect™ Real-Time PCR Detection System (Bio-Rad). The following primer pairs were used to amplify the target genes: *ARFa8* (PML399, 400), *ARFa6* (PML409, 419), *RSL4*; Pp1s164_82V6.1 (PML614, 615), *GCN5*; Pp1s20_204V6.1 (PML618, 619), *SAUR*; Pp1s4_222V6.1 (PML624, 625), *AUX1*; Pp1s56_28V6.1 (PML626, 627), *CBS domain*; Pp1s11_325V6.1 (PML810, 811), *Dox*; Pp1s15_259V6.1 (PML812, 813), *Epimerase*; Pp1s189_34V6.3 (PML814, 815), *bHLH TF*; Pp1s231_17V6.1 (PML818, 819), *Ubiquitin Ligase*; Pp1s37_28V6 (PML822, 823), *IAA1A* (IAA1A-F', IAA1A-R'), *IAA1B* (IAA1B-F', IAA1B-R') *IAA2* (IAA2-F', IAA2-R'), *EF1α*; Pp1s84_186V6.1 (EF1α-F', EF1α-R'). The sequences are listed in *Supplementary file 3*. Normalized expression (ΔΔC(t) method) was calculated using the Bio-Rad CFX manager software using *PpEF1α* as a reference gene and plotted as relative values ± SEM. Each analysis included three biological and four technical replicates. qPT-PCR analysis comparing gene expression levels between *arfb2 arfb4Δ aux/iaaΔ* and *aux/iaaΔ* lines included four biological replicates.

## Electrophoresis mobility shift assay (EMSA)

cDNA fragments encoding ARFb4 and ARFa8 DNA binding domains were amplified using primer pairs PML 455 & 851 and PML 461 & 852 respectively, and cloned into pDEST15 (Invitrogen). Recombinant GST fusions were expressed in *Escherichia coli* strain BL21-AI (Invitrogen) and purified by GST-agarose affinity. The eluted proteins were dialyzed in 100 mM Tris pH7.5, 100mM KCl, 5 mM MgCl₂. Electrophoresis mobility shift assay was carried out using LightShift Chemiluminescent EMSA Kit (Pierce, 20148). DNA oligonucleotides were biotinylated using Biotin 3' End DNA Labeling Kit (Thermo, 89818), annealed and used as DNA probes. For each binding reaction 150 fmol of DNA probe was incubated with x protein at room temperature for 20 min in a final volume of 20 μl containing binding buffer (20 mM Tris pH 7.5, 100 mM KCl, 2 mM DTT, 5% glycerol, 0.1 NP40, 10 MgCl₂), 0.5 μg Poly (dI•dC), and in the presence or the absence of excess molar ratio of specific or mutated unlabeled DNA competitor. The resulting protein-DNA complexes were electrophoresed on 5% native polyacrylamide gels, and then transferred to a Hybond N+ nylon membrane (GE Healthcare). Biotin labeled DNA detection was carried out according to the LightShift Chemiluminescent EMSA Kit manufacturer's instructions.

| Gene promoter region | Specific DNA probe | Mutated oligonucleotide |
|---|---|---|
| *DR5* | PML 839, 840 | PML 841, 842 |
| *ARFb4* | PML 1108, 1109 | PML 1110, 1111 |
| *IAA1A* | PML 1082, 1083 | PML 1084, 1085 |
| *IAA1B* | PML 1068, 1069 | PML 1094, 1095 |

## Plant growth assay

Following protoplast recovery, cellophane overlays were transferred to BCD plates. Same plants were imaged every day for five days using a Nikon SMZ1500 dissecting scope for five days and their length was measured using ImageJ.

## Cell measurement

Following tissue homogenization, plants were grown for seven days on BCDAT plates with cellophane overlays and imaged by confocal microscope (Zeiss MLS 710). Cells proximal to filament branches were selected. The cell length and width of thirty cells from different confocal photos were measured using ImageJ.

## Chlorophyll measurement

Plant tissue grown for seven days on BCDAT plates with cellophane overlays was extracted in 80% Acetone for 24 hr. Light absorbance of chlorophyll a and b were measured at wavelengths 663 and 646 nm using spectrophotometer and total chlorophyll concentration was calculated.

## GUS staining

Tissue was stained in GUS staining solution (50 mM $NaH_2PO_4$ (pH7.0), 0.5 mM X-Gluc, 0.5 mM K3Fe(CN) 6, 0.5 mM K4Fe(CN)6, 0.05% Triton X-100) at 37°C for 5 hr following the auxin treatment of $P_{IAA1B}$:GUS background lines. Plants were cleared in 70% (v/v) ethanol and imaged using a Nikon SMZ1500 dissecting scope.

## Protein Immunoblot Analysis

Proteins from protonemal tissue were extracted in 65 mM Tris pH6.8, 2% SDS, and 10% glycerol. The resulting protein extract was centrifuged for 10 min at 10,000 g and the supernatant was collected. Proteins were resolved by SDS-PAGE and transferred onto nitrocellulose membranes. Membranes were stained with Ponceau-S to standardize the input and an HRP-conjugated monoclonal anti-c-Myc 9E10 (Roche) or anti-GR P-20 (Santa Cruz) antibodies were used for protein detection. Proteins were visualized using ECL Plus Western Blot Detection System (GE healthcare).

## Phylogenetic analysis

Full-length ARF protein sequences were extracted from the *Physcomitrella patens, Selaginella moellendorffii*, and *Arabidopsis thaliana* protein databases downloaded from the Joint Genome Institute plant genome website (http://www.phytozome.org, accessed 14 Jan, 2014). The sequences were aligned using T-Coffee (*Notredame et al., 2000*), and poorly aligned regions were removed from the alignment. The tree was inferred using MrBayes [v3.2.2 x64; (*Ronquist et al., 2012*)] with the following parameters: aamodelpr=mixed, nst = 6, rates = invgamma, nruns = 2, nchains = 4, and ngen = 2000000. The consensus tree was visualized and exported using FigTree (http://tree.bio.ed.ac.uk/software/figtree/). Gene identifiers are as follows:

AtARF1, At1g59750; AtARF2, At5g62000; AtARF3_ETT, At2g33860; AtARF4, At5g60450; AtARF5_MP, At1g19850; AtARF6, At1g30330; AtARF7_NPH4, At5g20730; AtARF8, At5g37020; AtARF9, At4g23980; AtARF10, At2g28350; AtARF11, At2g46530; AtARF12, At1g34310; AtARF13, At1g34170; AtARF14, At1g35540; AtARF15, At1g35520; AtARF16, At4g30080; AtARF17, At1g77850; AtARF18, At3g61830; AtARF19, At1g19220; AtARF20, At1g35240; AtARF21, At1g34410; AtARF22, At1g34390; AtARF23, At1g43950; PpARFa1, Pp3c1_14480V3.1/Pp1s86_1V5_1; PpARFa2, Pp3c1_14440V3.1/—; PpARFa3, Pp3c2_25890V3.1/Pp1s119_25V5_1; PpARFa4, Pp3c13_4720V3.1/Pp1s133_57V5_1; PpARFa5, Pp3c26_11550V3.1/Pp1s6_230V5_3; PpARFa6, Pp3c17_19900V3/Pp1s65_225V5 (edited); PpARFa7, Pp3c14_16990V3.15/Pp1s48_142V5_1; PpARFa8, Pp3c1_40270V3/Pp1s163_120V5_1 (edited); PpARFb1, Pp3c27_60V3.1/Pp1s280_7V5_1; PpARFb2, Pp3c16_6100V3.1/Pp1s341_4V5_1; PpARFb3, Pp3c5_9420V3.1/Pp1s64_136V5_1; PpARFb4, Pp3c6_21370V3.1/Pp1s14_378V5_1; PpARFc1A, Pp3c4_12970V3.1/Pp1s339_45V6_1; PpARFc1B, Pp3c4_13010V3.1/—; PpARFc2, Pp3c6_26890V3.1/Pp1s279_8V5_1; PpARFd1, Pp3c9_21330V3.1/Pp1s316_22V6_1; PpARFd2, Pp3c15_9710V3.1/Pp1s250_62V6_1;

SmARFa1, Selmo117217 (edited); SmARFa2, Selmo424114 (edited); SmARFa3, Selmo181406 (edited); SmARFb1, Selmo437944 (edited); SmARFb2, Selmo81992 (edited); SmARFc1, Selmo61688 (edited); SmARFc2, Selmo51695 (edited); SmARFd1, Selmo1d28604 (edited); and SmARFd2, Selmo1d115320 (edited).

## Acknowledgements

We thank Daniel Lang for analyzing a related RNAseq data set prior to this work and to Ralph Quatrano for sharing materials. Work in the author's lab was supported by grants from NIH (GM43644 to ME), the Gordon and Betty Moore Foundation (to ME) and the Howard Hughes Medical Institute.

## Additional information

### Funding

| Funder | Grant reference number | Author |
|---|---|---|
| National Institute of General Medical Sciences | GM43644 | Mark Estelle |
| Howard Hughes Medical Institute | | Mark Estelle |
| Gordon and Betty Moore Foundation | | Mark Estelle |

The funders had no role in study design, data collection and interpretation, or the decision to submit the work for publication.

### Author contributions

ML, Designed the experiments, Performed the experiments, Analysis and interpretation of data, Wrote the manuscript, Conception and design, Drafting or revising the article; MJP, Designed the experiments, Performed the experiments, Analysis and interpretation of data, Conception and design, Drafting or revising the article; ST, Performed the experiments, Conception and design, Analysis and interpretation of data, Drafting or revising the article; SS, Performed the experiments, Drafting or revising the article; AK, Performed the experiments, Analysis and interpretation of data, Drafting or revising the article; KK, Performed the experiments, Conception and design, Drafting or revising the article, Contributed unpublished essential data or reagents; ME, Designed the experiments, Analysis and interpretation of data, Wrote the manuscript, Conception and design, Drafting or revising the article

### Author ORCIDs

Mark Estelle, http://orcid.org/0000-0002-2613-8652

## Additional files

### Supplementary files

• Supplementary file 1. (A) Differentially expressed genes between mock- and IAA treated WT protonemata, (fold change ≥1.5; padj <0.01). (B) Differentially expressed genes between mock treated WT protonemata and the *aux/iaaΔ* mutant (fold change ≥1.5; padj <0.01). (C) Differentially expressed genes between IAA treated WT protonemata, and the *aux/iaaΔ* mutant (fold change ≥1.5; padj <0.01). (D) Differentially expressed genes between mock- and IAA treated *aux/iaaΔ* mutant (padj <0.05).

• Supplementary file 2. (A) GO terms of biological processes (P), molecular function (F), and cellular component (C) associated with auxin-regulated genes. (B) GO terms of biological processes (P), molecular function (F), and cellular component (C) associated with genes regulated between WT and the *aux/iaaΔ* mutant.

• Supplementary file 3. Primer sequences used in this study.

• Supplementary file 4. Backbone vectors used in this study.

## Major datasets

The following dataset was generated:

| Author(s) | Year | Dataset title | Dataset URL | Database, license, and accessibility information |
|---|---|---|---|---|
| Meirav Lavy, Mark Estelle | 2016 | Physcomitrella patens Raw Sequence Reads | http://www.ncbi.nlm.nih.gov/bioproject/?term=PRJNA317343 | Publicly available at NCBI BioSample (accession no: PRJNA317343) |

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
