## [Decision Letter]

Thank you for submitting your work entitled "Constitutive auxin response in *Physcomitrella* reveals complex interactions between the Aux/IAA and ARF proteins" for consideration by *eLife*. Your article has been favorably evaluated by Detlef Weigel (Senior editor) and three reviewers, one of whom is a member of our Board of Reviewing Editors.

The reviewers have discussed the reviews with one another and the Reviewing Editor has drafted this decision to help you prepare a revised submission. All reviewers agree that your work represents an important advance in the field. However, they have a number of concerns that have to be addressed before publication. This includes the following revisions, which we consider essential:

1) Please provide a better characterization of the 7-day-old protonema from wild type and the mutant, at the morphological and physiological level, to clarify to what degree observed resting state expression differences could reflect secondary effects.

2) Please show that ARF activators are really ARE activators and that ARF repressors are really ARE repressors in a direct assay, not only based on phylogeny. For example, the '"classic" protoplast assays to monitor ARF activity could be helpful here.

3) Please demonstrate that different ARFs can bind to or compete for the same binding site. This could be done by ChIP experiments, or by gel-shift experiments, using a few example genes and appropriate controls.

4) Please show experimentally that the *aux/iaa* null is insensitive to depletion of auxin levels (e.g. with L-kyn), and that the *ARFB4OE* line 1 (shown in Figure 3) does not produce caulonema if grown on NAA.

5) Please fix the formatting errors that are spread throughout the manuscript and consider a rewrite of some sections to make the paper easily accessible for the reader.

Beyond these essential revisions, the reviewers brought up a number of additional points that you should ideally address as well. They are listed per reviewer below:

*Reviewer #1:*

This paper from the Estelle lab attacks the question of auxin signaling redundancy through a simple biological system, *Physcomitrella*. The work is surely interesting and by a large, the experiments appear to support the claims. There is ample room for improvement, however.

To start with, I feel the paper is overall not that well written and would benefit from a thorough rephrasing in a number of places. As it is written now, the paper is sometimes hard to follow. There are also various formatting issues that have to be fixed (example given, order of figure panel reference, panel organization, reference to inexistent figure panels like 3D).

The authors present a *Physco* mutant in which they knocked out all three *Aux/IAA* genes. Given the wide implications of their results, I think it would be pertinent to check this line in depth by whole genome sequencing, to definitely exclude the presence of any residual Aux/IAA activity, for instance because of strain variation.

When the authors say that the *aux/iaa* mutant filaments did not display normal pattern of differentiation, could they be more specific? From the macroscopic images, the mutant sometimes just looks like a small wild type (Figure 1 right panel) to someone not familiar with the *Physco* system.

I believe an important addition to the paper would be a more detailed, microscopic analysis of the protonema stage in wild type and the mutant. The authors find a large number of gene expression differences in the *aux/iaa* mutant, with ca. 80% of those genes not being auxin-responsive in wild type. I believe this does not really mean that these genes are directly regulated by the AUX/IAA-ARF system. Presumably, many of those differences could arise as secondary effects, for instance because after all, the protonema grown from wild type and mutant is not that similar morphologically as well as physiologically. Differences could also be amplified by the phenotypic variation the authors describe. So I believe they have to be more careful in their claims.

Regarding the ARF analysis, I believe it would considerably strengthen the manuscript if the authors could provide RNAseq data for the mutant compared to the mutant carrying the *ARFb4OE* construct.

*Reviewer #2:*

The manuscript by Lavy et al. addresses the function of auxin response components in *Physcomitrella*. Little is known about the evolution of auxin signaling, or its ancestral role, and this manuscript fills in many gaps. By studying a plant lacking all Aux/IAA function, the authors show the impact of auxin response on the genome, which turns out to be very large. This, in itself is a very important finding. Next, by functionally studying several ARFs, the authors come up with a model of gene regulation by functionally distinct ARF types. The manuscript is generally of good quality and will have major impact in the field. There are however a number of rough edges and lacking controls, as well as leaps of logic that I would like to see addressed.

1) The transcriptome shows large effects of Aux/IAA deletion on gene expression. However, the analysis was done in 7-day old protonema, which allows for numerous layers of secondary effects. While this is not necessarily a problem, it becomes difficult when the authors use this approach to support their claim that "activating and repressing ARFs share target genes". To make such a strong and direct statement, other types of support are necessary, such as for example EMSA or ChIP.

2) An alternative interpretation to the unexpected requirement of "repressive" ARFs for auxin-dependent gene activation is that the two types of ARFs interact, and that heterodimers of both mediate gene activation *and* repression. Yet another interpretation is that the distinction between the function of activating and repressing ARFs is not strict, and the ARF2 and ARF4 also contribute to gene activation. Similarly, ARF5 in *Arabidopsis* has been shown to repress certain target genes (ARR7, ARR15 and STOMAGEN). I would ask the authors to discuss this and consider multiple interpretations.

3) A central premise of the current manuscript is based on the distinction between activating and repressing ARFs. However, the definition seems to be based only on phylogenetic relationship to the few *Arabidopsis* ARFs that were functionally assayed. However, a lot of time has passed since *Physcomitrella* and the precursor of *Arabidopsis* diverged. The authors seem to rule out the possibility that functions have changed in the past few 100's of millions of years. I would suggest that the assumptions about being activators or repressors are tested according to standard assays (using synthetic promoter in a protoplast assay), at least for the ARFs used here (ARFb2, b4, a8).

*Reviewer #3:*

This paper presents an impressive body of work that yields fundamental new insights into auxin signaling – a key regulatory process in plant development. As such I think that it will be an important contribution to the literature. I found the genetics quite heavy duty, but I think that the work is presented in a clear and logical way, and I thought that most of the conclusions drawn were fully supported by the data shown.

The only conclusion that I thought should be strengthened is to do with the auxin blindness of the *aux/iaa* null – it is conceivable that the mutant has very high auxin levels, and if it has a maxed-out auxin response, I would not expect it to be sensitive to the addition of further auxin (Figure 1).

I would like to see that the *aux/iaa* null is insensitive to depletion of auxin levels (e.g. with L-kyn), and that *ARFB4OE* line 1 (shown in Figure 3) does not produce caulonema if grown on NAA.

I would like to see the data showing that the *aux/iaa* null is non responsive to auxin in Figure 2.

---

## [Author Response]

All reviewers agree that your work represents an important advance in the field. However, they have a number of concerns that have to be addressed before publication. This includes the following revisions, which we consider essential:

*1) Please provide a better characterization of the 7-day-old protonema from wild type and the mutant, at the morphological and physiological level, to clarify to what degree observed resting state expression differences could reflect secondary effects.*

A) We now present a more detailed description of the growth conditions and the experiment setup of the RNAseq experiment performed on 7-day-old plants. The main point of the extended description was to show that although the phenotype of the *aux/iaa* mutant is very severe, we performed the experiment under conditions that minimize the differences between WT and the mutant, as much as possible. Using these conditions we were also able to collect healthy mutant tissue.

B) We now include a characterization of wild-type and mutant plants grown under the same specific conditions as for the RNAseq. This includes the microscopic analysis of the cells including measurements of length and width, as well as measurements of chlorophyll concentration (Figure 2, Figure 2—figure supplement 1).

C) In addition to the data described above, we now also present a clearer analysis of 7-day-old plants following protoplast recovery in a new supplemental figure (Figure 1—figure supplement 2). This was done in order to compare growth rate and individual plant morphology, which was not possible using the conditions adopted for RNAseq (in the case of the RNAseq we grew plants from blended tissue in order to be able to collect the same amount of tissue).

D) Finally, we emphasized in the text that differentially expressed genes in the mutant will include indirect auxin targets.

2) Please show that ARF activators are really ARE activators and that ARF repressors are really ARE repressors in a direct assay, not only based on phylogeny. For example, the '"classic" protoplast assays to monitor ARF activity could be helpful here.

Unfortunately we were unable to obtain conclusive results using several transient expression assays based on moss or *Arabidopsis* protoplasts. However, we strongly believe that our conclusions not affected by the absence of this data for three reasons.

A) In a recent paper describing the auxin pathway in *Marchantia polymorpha* (Kato et al., 2015, Plos Genetics) the authors confirm that MpARF2 predicted to be repressing ARFs based on phylogeny, did indeed act to repress transcription in tobacco cells. Since *Marchantia* is an early diverged land plant, these results strongly suggest that ARF function is conserved in all land plants.

B) In Plavskin et al., (Dev Cell 36 276-289), the authors show that accumulation of PpARFbs (in a Ppsgs3 mutant and in an over-expression line) results in reduces expression of auxin resistant genes and a morphology very similar to previously described auxin-resistant mutants.

C) Our genetic analysis of stable ARF lines clearly shows that different ARFs behave according to their phylogenetic classification. For example: in Figure 3 and Figure 5 we show that overexpression of repressing ARF in the *aux/iaa* mutant results in repression of gene expression. Inducible expression of an activating ARF in the same background line and results in opposite direction of gene expression levels as well as suppresses the repressing ARF phenotype.

We have added several sentences to the Results describing points A and B.

3) Please demonstrate that different ARFs can bind to or compete for the same binding site. This could be done by ChIP experiments, or by gel-shift experiments, using a few example genes and appropriate controls.

We performed gel shift experiments and presented the binding of ARF4b and ARFa8 to the same DNA sequencesin three auxin responsive gene promoters, in the absence and presence of specific or mutated competitors (Figure 4).

4) Please show experimentally that the aux/iaa null is insensitive to depletion of auxin levels (e.g. with L-kyn), and that the ARFB4OE line 1 (shown in Figure 3) does not produce caulonema if grown on NAA.

A) This is a good suggestion. We grew both WT and *aux/iaaΔ* mutant on L-kyn (Figure 1) and, as expected, differentiation of caulonema was inhibited by the compound in wild-type plants. In contrast the triple mutant was insensitive.

B) We now present data showing that even after exposure to NAA for one month, the *ARFB4OE (aux/iaaΔ)* line does not produce caulonema (Figure 3).

5) Please fix the formatting errors that are spread throughout the manuscript and consider a rewrite of some sections to make the paper easily accessible for the reader.

We apologize for these errors. All references to figures and panels have been carefully checked and corrected where needed. In addition, the order of panels and their references in the text was changed for clarity. Further, the first part of the manuscript was rewritten and some figures were reorganized to make the work more accessible for readers.

*Beyond these essential revisions, the reviewers brought up a number of additional points that you should ideally address as well. They are listed per reviewer below:*

*Reviewer #1:*

This paper from the Estelle lab attacks the question of auxin signaling redundancy through a simple biological system, Physcomitrella. The work is surely interesting and by a large, the experiments appear to support the claims. There is ample room for improvement, however.

*To start with, I feel the paper is overall not that well written and would benefit from a thorough rephrasing in a number of places. As it is written now, the paper is sometimes hard to follow. There are also various formatting issues that have to be fixed (example given, order of figure panel reference, panel organization, reference to inexistent figure panels like 3D).*

Please see our response to general revision comment #5.

*The authors present a Physco mutant in which they knocked out all three Aux/IAA genes. Given the wide implications of their results, I think it would be pertinent to check this line in depth by whole genome sequencing, to definitely exclude the presence of any residual Aux/IAA activity, for instance because of strain variation.*

We are not sure that we understand the reviewer’s concern. Is the reviewer proposing that there is a fourth *Aux/IAA* gene in our strain? If so, we have performed many PCR and RT-PCR experiments, and sequenced many PCR products, and there is no evidence of another gene. In addition, we have a number of lines of evidence (PCR, RT-PCR and RNAseq), all of which indicate the complete absence of *Aux/IAA* transcripts. This is consistent with the way the strain was generated such that all three genes are completely deleted. Finally, it is clear that the triple mutant completely lacks an auxin response.

When the authors say that the aux/iaa mutant filaments did not display normal pattern of differentiation, could they be more specific? From the macroscopic images, the mutant sometimes just looks like a small wild type (Figure 1 right panel) to someone not familiar with the Physco system.

As detailed in our response to general comment #1, we have made a number of changes to address this concern. To clarify the phenotypic analysis we have rewritten the pertinent section of the manuscript. In addition Figure 1 has been reorganized and two supplemental figures have been added (Figure 1—figure supplement 2 and Figure 2—figure supplement 1). In general, we separated the analysis of mature plants from the analysis of young plants and described each one in more detail.

I believe an important addition to the paper would be a more detailed, microscopic analysis of the protonema stage in wild type and the mutant. The authors find a large number of gene expression differences in the aux/iaa mutant, with ca. 80% of those genes not being auxin-responsive in wild type. I believe this does not really mean that these genes are directly regulated by the AUX/IAA-ARF system. Presumably, many of those differences could arise as secondary effects, for instance because after all, the protonema grown from wild type and mutant is not that similar morphologically as well as physiologically. Differences could also be amplified by the phenotypic variation the authors describe. So I believe they have to be more careful in their claims.

Please see our response to general revision comment #1.

Regarding the ARF analysis, I believe it would considerably strengthen the manuscript if the authors could provide RNAseq data for the mutant compared to the mutant carrying the ARFb4OE construct.

We agree that this would be an interesting data set. However, in our opinion, the RT-PCR data that we present clearly show the effects of ARFb4 over-expression, together with the phenotype of these plants.

Reviewer #2:

*The manuscript by Lavy et al. addresses the function of auxin response components in Physcomitrella. Little is known about the evolution of auxin signaling, or its ancestral role, and this manuscript fills in many gaps. By studying a plant lacking all Aux/IAA function, the authors show the impact of auxin response on the genome, which turns out to be very large. This, in itself is a very important finding. Next, by functionally studying several ARFs, the authors come up with a model of gene regulation by functionally distinct ARF types. The manuscript is generally of good quality and will have major impact in the field. There are however a number of rough edges and lacking controls, as well as leaps of logic that I would like to see addressed. 1) The transcriptome shows large effects of Aux/IAA deletion on gene expression. However, the analysis was done in 7-day old protonema, which allows for numerous layers of secondary effects. While this is not necessarily a problem, it becomes difficult when the authors use this approach to support their claim that "activating and repressing ARFs share target genes". To make such a strong and direct statement, other types of support are necessary, such as for example EMSA or ChIP.*

Please see our response to general revision comment #3.

2) An alternative interpretation to the unexpected requirement of "repressive" ARFs for auxin-dependent gene activation is that the two types of ARFs interact, and that heterodimers of both mediate gene activation and repression. Yet another interpretation is that the distinction between the function of activating and repressing ARFs is not strict, and the ARF2 and ARF4 also contribute to gene activation. Similarly, ARF5 in Arabidopsis has been shown to repress certain target genes (ARR7, ARR15 and STOMAGEN). I would ask the authors to discuss this and consider multiple interpretations.

The discussion regarding possible mechanisms of ARF activity has been expanded to include more possible interpretations (third paragraph in the Discussion).

3) A central premise of the current manuscript is based on the distinction between activating and repressing ARFs. However, the definition seems to be based only on phylogenetic relationship to the few Arabidopsis ARFs that were functionally assayed. However, a lot of time has passed since Physcomitrella and the precursor of Arabidopsis diverged. The authors seem to rule out the possibility that functions have changed in the past few 100's of millions of years. I would suggest that the assumptions about being activators or repressors are tested according to standard assays (using synthetic promoter in a protoplast assay), at least for the ARFs used here (ARFb2, b4, a8).

Please see our response to general revision comment #2.

Reviewer #3:

This paper presents an impressive body of work that yields fundamental new insights into auxin signaling – a key regulatory process in plant development. As such I think that it will be an important contribution to the literature. I found the genetics quite heavy duty, but I think that the work is presented in a clear and logical way, and I thought that most of the conclusions drawn were fully supported by the data shown.

The only conclusion that I thought should be strengthened is to do with the auxin blindness of the aux/iaa null – it is conceivable that the mutant has very high auxin levels, and if it has a maxed-out auxin response, I would not expect it to be sensitive to the addition of further auxin (Figure 1).

*I would like to see that the aux/iaa null is insensitive to depletion of auxin levels (e.g. with L-kyn), and that ARFB4OE line 1 (shown in Figure 3) does not produce caulonema if grown on NAA.*

Please see our response to general revision comment #4.

I would like to see the data showing that the aux/iaa null is non responsive to auxin in Figure 2.

For the hierarchical clustering shown in that panel (now 2E) only genes with significant differential expression between untreated and treated WT were selected. This gene set was than compared with the mutant gene set (we chose to present the untreated mutant gene set, but to the same purpose we could use the treated gene set). Since there are zero genes that present significant differential expression (p<0.01) between the treated and untreated mutant, adding a fourth column (treated mutant) is statistically wrong, as it would include a comparison between genes with insignificant differential expression. To emphasize the mutant’s insensitivity, and since under our statistical conditions there is no data to show, we presented a table ([Supplementary-material SD1-data]) showing that even under a less stringent statistical threshold only a few genes are present. In order to include a visual representation for the mutant’s auxin insensitivity, we now have added another circle to the Venn diagram in Figure 2 illustrating the absence of differential expressed genes between the mutant gene sets under our statistical thresholds.